# STINGing Defenses: Unmasking the Mechanisms of DNA Oncovirus-Mediated Immune Escape

**DOI:** 10.3390/v16040574

**Published:** 2024-04-09

**Authors:** Mayra F Martínez-López, Claire Muslin, Nikolaos C. Kyriakidis

**Affiliations:** 1Cancer Research Group (CRG), Faculty of Medicine, Universidad de las Américas, Quito 170503, Ecuador; mayra.martinez.lopez@udla.edu.ec; 2One Health Research Group, Faculty of Health Sciences, Universidad de las Américas, Quito 170503, Ecuador; claire.muslin@gmail.com

**Keywords:** STING, DNA oncovirus, cancer, innate immunity, immune evasion, cancer therapy

## Abstract

DNA oncoviruses represent an intriguing subject due to their involvement in oncogenesis. These viruses have evolved mechanisms to manipulate the host immune response, facilitating their persistence and actively contributing to carcinogenic processes. This paper describes the complex interactions between DNA oncoviruses and the innate immune system, with a particular emphasis on the cGAS-STING pathway. Exploring these interactions highlights that DNA oncoviruses strategically target and subvert this pathway, exploiting its vulnerabilities for their own survival and proliferation within the host. Understanding these interactions lays the foundation for identifying potential therapeutic interventions. Herein, we sought to contribute to the ongoing efforts in advancing our understanding of the innate immune system in oncoviral pathogenesis.

## 1. Introduction

The intricate landscape of oncogenesis, the process by which normal cells transform into cancerous entities, remains a focal point of scientific inquiry due to its profound implications for human health. While genetic mutations and environmental factors are recognized as major drivers of carcinogenesis, research also underscores the significant contribution of DNA viruses in this complex biological phenomenon.

### 1.1. Oncogenic DNA Viruses

Approximately 10% of human cancer cases worldwide are attributed to viral infections [1]. Among the seven recognized human oncoviruses, five are DNA viruses: Epstein–Barr virus (EBV), Kaposi sarcoma-associated herpesvirus (KSHV), Human Papillomavirus (HPV), Hepatitis B virus (HBV) and Merkel Cell Polyomavirus (MCPyV) (Table 1) [2]. Furthermore, oncogenic viruses are widespread among domestic animals, causing significant morbidity, mortality and economic losses. Among virus-induced cancers of utmost veterinary significance, both bovine papillomatosis and poultry Marek’s disease are caused by DNA viruses [3].

Oncoviral tumorigenesis is a complex multistep process in which viral infection appears to be necessary but insufficient. Many additional events, such as chronic inflammation, environmental mutagens, or immunosuppression synergize viral infection to malignant transformation [4]. A common feature of DNA oncoviruses is their ability to establish persistent infections, characterized by reduced or absent productive virus replication. During viral persistence, the virus hides from the immune system by turning off the expression of unnecessary viral proteins that might be sensed by cell-mediated immune recognition mechanisms, thus preventing viral elimination. The virus persists as a naked DNA genome, often as a plasmid or episome, which relies on host cell machinery to replicate whenever the cell divides [5]. Therefore, to support viral propagation, the genomes of oncoviruses typically encode oncoproteins that target tumor suppressor pathways, apoptotic signaling or host DNA damage response to force unscheduled S phase entry and cell proliferation [6]. By targeting the cell cycle checkpoints and anti-apoptotic machinery that are involved in genomic proofreading, viral oncoproteins also induce cellular genomic instability and aneuploidy, which in turn contribute to carcinogenesis. In particular, integration of the viral genome into host chromosomes has been shown to be a causal mechanism in several cancers [7,8,9].

Additionally, viral oncoproteins frequently manipulate host immune responses, including sensing of pathogen-associated molecular patterns (PAMPs), immune gene expression profiles and intercellular signaling, to evade detection and elimination during both primary infection and persistency. Viral immune evasion has been suggested to potentiate cancer since the mechanisms employed to evade detection also impede the effective surveillance of transformed cells. Additionally, these mechanisms can accelerate cellular proliferation [10]. Thus, we will now delve into the key features of DNA oncoviruses, in particular, their role in neoplastic transformation in both humans and animals.

#### 1.1.1. Herpesviruses

Herpesviruses are large, enveloped viruses with a linear, double-stranded DNA genome of 120 to 240 kb. They belong to the *Orthoherpesviridae* family, which is divided into three subfamilies: *Alphaherpesvirinae*, *Betaherpesvirinae* and *Gammaherpesvirinae*. Herpesviruses are highly prevalent worldwide and capable of infecting a broad spectrum of vertebrates. They have generally coevolved with their hosts and are highly adapted to them [11]. Following primary infection, herpesviruses are able to persist in the host by evading host immune surveillance and establishing latency. Latently infected cells serve as a perpetual reservoir, enabling the amplification of progeny viruses for dissemination within the host and transmission between hosts. Thus, herpesviruses typically display a biphasic life cycle, alternating between lytic replication and latency. During the lytic productive replication cycle associated with primary infection and occasional reactivation, a broad range of viral proteins are involved in replicating the viral genome and producing new viral particles. The latent cycle, which enables the virus to persist long term in the infected host without producing viral particles, is characterized by minimal gene expression with chromatin-tethered viral episomes replicating and dividing in synchrony with the cell [12,13,14]. Members of the subfamilies *Alphaherpesvirinae*, *Betaherpesvirinae* and *Gammaherpesvirinae* establish latency in neuronal, myeloid and lymphoid cells, respectively [11].

Marek’s disease virus (MDV), or Gallid herpesvirus 2 (GaHV-2), belonging to the *Alphaherpesvirinae* subfamily, is an economically important virus that induces fatal T cell lymphomas in chickens. MDV infection starts with the inhalation of dander and dust containing infectious virus particles. After early cytolytic replication in macrophages and B cells, the virus establishes latent infection in CD4+ T cells, which can subsequently undergo transformation, resulting in the formation of deadly lymphomas in the skin and visceral organs [15]. Latently and/or productively infected T cells transport the virus to the skin and shedding occurs from feather follicle epithelial cells [16]. In latently infected cells, MDV integrates its viral genome into the telomeres of host chromosomes. This integration ensures the maintenance of the virus genome and is thought to be crucial for T cell transformation. Several viral genes involved in oncogenesis have been identified within the MDV genome. In particular, the Meq oncogene is known to play a key role in MDV-induced T cell lymphomagenesis, by impacting the expression of cellular anti-apoptotic factors and viral transformation-associated genes and binding to cell cycle control factors [17].

Two human gammaherpesviruses, EBV and KSHV, are known as causative agents for a variety of tumors. EBV is a ubiquitous DNA virus that persistently infects more than 90% of the human population [18]. EBV is transferred via saliva exchange and infects mucosal epithelial cells as well as B cells located in submucosal secondary lymphoid tissues. The virus eventually establishes a long-term infection in memory B cells [19]. Primary infection with EBV at an early age is mostly asymptomatic, but acquiring the virus during adolescence or later in life can lead to infectious mononucleosis [20]. Furthermore, EBV is estimated to cause 1–2% of all tumors in humans and approximately 200,000 new cancers per year [21]. Reflecting the cell types that EBV normally infects, EBV-associated cancers are mainly lymphomas derived from B cells and nasopharyngeal and gastric carcinomas [12]. EBV produces several oncogenic proteins that combine pro-proliferative and anti-apoptotic functions and that have been suggested to drive infected B cells through their activation into the germinal center and further differentiation into long-lived memory B cells. The main EBV oncoproteins identified include the six EBV nuclear antigens (EBNAs) and the two latent membrane proteins (LMPs). The EBV-activated germinal center differentiation of infected B cells is thought to lead to the acquisition of additional, growth-transforming mutations via the machinery that diversifies the B cell receptor [19].

KSHV, also known as human herpesvirus 8 (HHV-8), is responsible for Kaposi’s sarcoma (KS), the most common cancer among HIV-infected patients and among men in sub-Saharan Africa [22,23,24]. KSHV is also the cause of two other rare lymphoproliferative disorders, namely primary effusion lymphoma (PEL) and multicentric Castleman’s disease (MCD) [25,26]. KSHV shows a very unequal geographical distribution: its seroprevalence is estimated to be greater than 50% only in sub-Saharan Africa, intermediate (5–20%) around the Mediterranean basin and in some Latin America countries, and less than 5% in most other parts of the world [27,28]. Viral transmission can occur via saliva, blood, and sexual contact. KSHV displays a broad cellular tropism and has been detected in vivo in endothelial cells, epithelial cells, monocytes and B cells where it establishes a latent reservoir [29]. B lymphocytes are considered the primary site of viral latency, which can promote dissemination throughout the body. Latent infection of endothelial cells does not appear to establish a long-term viral reservoir but is the essential primary event giving rise to KS development [30]. Several of the limited viral proteins expressed in latently infected cells are thought to play an essential role in the pathogenesis of KSHV-associated malignancies by creating a favorable microenvironment for tumor initiation and progression [6]. In particular, the latency-associated nuclear antigen (LANA), the most consistently detected and abundantly expressed protein in all KSHV-infected tumor cells, has been found to block TGF-β signaling and inhibit p53, impairing apoptosis and increasing cell proliferation and survival [29]. The transition to the lytic phase, which is infrequent in vivo, results in the expression of all viral genes, including some that encode proteins that play pivotal roles in promoting oncogenesis. Therefore, both phases of the virus life cycle are thought to play significant roles in the pathogenesis of KS [31].

#### 1.1.2. Papillomaviruses

Papillomaviruses are small, nonenveloped double-stranded DNA viruses of the *Papillomaviridae* family, which have specific tropism for keratinocytes [32]. They have a circular genome of approximately 8 kb, comprising eight early open reading frames (ORFs), two late ORFs and a noncoding control region. Papillomaviruses are able to infect all vertebrates and are known to produce benign tumors such as warts on the skin and condylomas on mucous membranes in various species. Additionally, they can be associated with the development of epithelial malignancies and cancer [33,34,35]. HPV is the most common sexually transmitted viral infection worldwide [36,37]. It has been estimated that, in the United States, more than 80% of sexually active women and men will acquire at least one HPV infection by the age of 45 years [38]. In women, 90% of incident HPV genital infections clear within two years without any clinical impact [39]. However, when persistence occurs, HPV can become a risk factor for malignant transformation. Over 200 HPV types have been identified, that can be classified into two categories: low-risk HPVs responsible for anogenital and cutaneous warts, and high-risk HPVs responsible for cervical, anogenital, and oropharyngeal cancers [40,41,42]. With an estimated 604,000 new cases and 342,000 deaths worldwide in 2020, cervical cancer ranks fourth in cancer incidence and mortality among women [43]. The 12 high-risk HPVs that have been identified are responsible for virtually all cervical cancers, and among these, HPV16 and 18 are the most virulent high-risk genotypes, causing about 70% of invasive cervical cancer in the world [44,45,46].

The HPV replication cycle and viral gene expression are dependent on epithelial differentiation [47,48]. HPVs infect the basal stem cells of the stratified squamous cutaneous and mucosal epithelia, which are the only actively dividing epithelial cells. A successful infection requires a lesion in the stratified tissue, providing access to the basal layer of the epithelium [49]. Following entry into basal keratinocytes, HPV must wait for mitosis in order for the viral DNA genome to enter the nucleus and establish as an episome [50]. Subsequently, transcription of early ORFs can occur, mediated by host cellular factors. During the division of infected basal cells, HPV episomes are replicated together with cellular chromosomes and distributed equally to the new basal cell and the daughter cell that will undergo differentiation. As this infected daughter cell differentiate and migrate towards the surface of the tissue, the viral genomes are replicated and structural proteins are expressed, allowing virion assembly and release from the top layer of the differentiated epithelium [42,51].

Cancer progression is a rare event associated with persistent high-risk HPV infection and dependent on viral E6 and E7 oncoproteins [52]. HPV E6 and E7 proteins inactivate the p53 tumor-suppressor protein and the retinoblastoma protein, respectively, thereby driving cell cycle entry and cell proliferation in the basal and parabasal epithelial cell layers [42,53]. The trigger for cell transformation is thought to be the integration of viral episomes into the host genome, which leads to deregulation of E6 and E7 expression and subsequent excessive cell proliferation, deficient DNA repair, and the accumulation of genetic damage in the infected cell [44,54].

BPV infections have been observed in cattle in multiple locations worldwide, exerting significant impacts on livestock production and local economies [55]. Currently, 29 BPV types have been characterized and classified into five genera: *Delta*, *Xi*, *Epsilon*, *Dyoxi* and *Dyokappapapillomavirus* [56]. BPVs are typically responsible for persistent infections, and most BPV types are associated with benign cutaneous papillomas and fibropapilomas that often regress spontaneously [57]. However, four BPV types, namely BPV-1, -2, -4 and -13, are considered highly pathogenic and frequently associated with cutaneous and mucosal tumors in cattle. Deltapapillomaviruses BPV-1, BPV-2 and BPV-13 are notably responsible for epithelial and mesenchymal urinary bladder tumors in both cattle and buffaloes [58,59,60]. Additionally, BPV-4, from the *Xipapillomavirus* genus, is associated with papillomas and squamous cell carcinomas in the upper digestive tract, including the oral and pharyngeal cavities, the esophagus and the rumen [57]. The major oncoprotein of BPV, E5, a small membrane-associated protein, demonstrates transforming activity through various pathways, notably promoting cell proliferation via activation of the platelet-derived growth factor receptor beta (PDGFRβ) [61]. Furthermore, clinical, epidemiological and experimental evidence highlights the significant role of exposure to bracken ferns (*Pteridium* spp.) in facilitating viral persistence and the malignant transformation of early viral lesions. Bracken illudane glycoside compounds, such as ptaquiloside, exhibit immune suppressive and mutagenic properties and act synergistically with BPVs, resulting in neoplastic disease [57].

#### 1.1.3. Hepatitis B Virus

In 2019, an estimated 296 million people worldwide were living with chronic HBV infection, and approximately 1.5 million people newly acquire HBV infection each year, despite the existence of a highly efficacious vaccine [62]. With over 800,000 annual deaths attributed to HBV-related liver cirrhosis and hepatocellular carcinoma, HBV continues to pose a significant global public health burden. HBV is a hepatotropic, non-cytopathic virus, a member of the *Hepadnaviridae* family. The small, enveloped virion contains a 3.2 kb, partially double-stranded, relaxed circular (rc) DNA genome that, unlike other DNA viruses, replicates via reverse transcription of an RNA intermediate, the pregenomic RNA (pgRNA) [63]. Upon infection, the rc DNA genome is converted to episomal covalently closed circular DNA (cccDNA) in the nucleus of human hepatocytes, where it serves as a template for viral RNA transcription and as a persistence reservoir. The core proteins package viral pgRNA and form the nucleocapsid. Viral reverse transcriptase then converts the pgRNA to the rcDNA genome and the matured nucleocapsid either is enveloped and secreted out of the cell, or delivers the rcDNA into the nucleus to amplify the pool of cccDNA [64,65].

The likelihood of HBV persistence strongly depends on age at acquisition. Perinatal infection from a carrier mother results in chronicity in most cases, whereas horizontal transmission during adulthood, such as through sexual exposure or the sharing of drug-injection needles, often leads to self-limited acute infection [66,67]. Upon infection, HBV induces little or no innate immune responses in the hepatocytes but is thought to be detected by non-parenchymal innate immune cells in the liver at a later incubation stage [66]. During self-limited acute hepatitis B, HBV is controlled, without being completely eliminated, by HBV-specific CD4+ and CD8+ T cell responses as well as neutralizing antibodies [68,69]. Inversely, the failure of HBV control and subsequent establishment of chronic infection is attributed to a defective adaptive immune response, influenced by both host and viral factors [70].

Liver disease is triggered by HBV-specific CD8+ T cells, both directly by killing infected hepatocytes and indirectly by recruiting pathogenic inflammatory cells into the liver [71]. Complications of chronic hepatitis B (CHB) usually take place after decades of low-level CD8+ T cell-dependent liver disease, where the coexistence of hepatocellular necrosis, hepatocellular regeneration and liver inflammation is believed to trigger abnormal repair functions and random genetic damage, ultimately leading to liver fibrosis, liver cirrhosis and hepatocellular carcinoma [71,72]. Additionally, HBV can directly promote carcinogenesis by three different mechanisms: (i) insertional mutagenesis with the integration of HBV DNA into cellular protooncogenes; (ii) promotion of chromosomal instability as the result of both the integration of viral DNA into the host genome and the activity of viral proteins; (iii) oncogenic activities of wild-type and mutated/truncated viral proteins that affect cell functions, including cell proliferation and cell viability, and sensitize liver cells to mutagens [73]. In particular, the HBV X protein (HBx) is thought to play an important role in HBV oncogenicity by directly or indirectly regulating the transcription of several genes involved in regulating DNA repair, expression of miRNAs, autophagy, cell proliferation and invasion, cell cycle progression and angiogenesis [6].

Overall, oncogenic DNA viruses have evolved various strategies to evade host immunity and establish persistent infections, thereby providing time for the processes underlying neoplastic transformation to occur.

### 1.2. Cancer Immunosurveillance

At the beginning of the 20th century, Paul Ehlrich hypothesized that there should be a host defense that prevents aberrant cells from progressing into tumors, otherwise, neoplastic malignancies would be commonly found in the human population [74]. In agreement with this initial hypothesis, it was later proposed that all long-lived organisms must have defense mechanisms against neoplasia [75]. Additionally, it was suggested that tumor cells could possess specific neoantigens with the capacity to stimulate the immune system. In line with this, it was observed that the incidence of cancer was higher in immunocompromised individuals [76]. These theories were later confirmed with functional experiments in mice, which showed that tumors transplanted among syngeneic hosts were rejected whereas other tissues were not [77]. These results suggested that the tumors must have specific antigens that would induce an immune response in these otherwise genetically identical hosts. As the knowledge of cancer etiopathology and immunology advanced, the role of the immune system in cancer development began to surface and the cancer immune surveillance hypothesis emerged.

This hypothesis posits that immune cells act as sentinels capable of detecting, targeting, and eliminating abnormal cells that have been compromised by mutations or other genetic alterations. Immune cells closely patrol all tissues, employing a range of mechanisms to detect changes in the expression patterns of membrane molecules as well as the presence of intracellular elements outside the cell [78,79,80]. A notable example of such changes is the loss of the major histocompatibility complex class I (MHC-I) in several types of tumors. Loss of MHC-I molecules is detected by natural killer (NK) cells, which subsequently release their cytotoxic response, culminating in the apoptosis of the mutated cell [81].

Due to its inherent molecular aberrations, the structure of tumors is defective and promotes cancer cell necrosis [82]. Necrotic material, such as double-strand DNA (dsDNA), is released, internalized, and readily detected by the immune system. Additionally, other sources of cytosolic dsDNA include nuclear, mitochondrial, and intracellular pathogens. Cytosolic dsDNA is recognized by several immune sensors such as TLR9, DNA-dependent protein kinase, RNA polymerase III, DEAD box polypeptide 41 (DDX41), PYHIN protein family, AIM-2-like receptors (ALRs) and, importantly the cyclic GMP-AMP synthase (cGAS)—stimulator of interferon genes (STING) pathway [83]. Among the viruses that activate the cGAS-STING pathway, there are several oncoviruses from the *Papillomaviridae* and *Herpesviridae* families, which were previously mentioned [84].

The constant selective pressure exerted by the immune system upon oncoviruses and pre-malignant cells has resulted in the development of immune evasion mechanisms. Both can hijack innate immune surveillance processes in order to create a tumor-permissive immune suppressive microenvironment [85,86]. Delineating the dual role that immune responses play in oncoviral-mediated malignant transformation is crucial for the development of new therapies.

### 1.3. STING Pathway

Innate immunity is the first line of defense against pathogens. Innate immune cells express on their cell surface, within the cytosol or even within endosomes pattern recognition receptors (PRRs) that allow for the recognition of PAMPs and host damage-associated molecular patterns (DAMPs). The presence of DNA (foreign or self) within the cytosol is recognized as a danger signal and potential sign of pathogen invasion. DNA can be brought into the cytosol through viral or bacterial infection. Additionally, the damage to nuclear or mitochondrial integrity can induce the release of DNA into cytosol. Cytosolic DNA is recognized by the cGAS-STING pathway that acts as a DNA sensor activating the innate immune system and the release of molecules like the type I interferon (IFN) [87,88,89,90]. Thus, it is important to understand the molecular underpinnings of this pathway in the study of immune evasion mechanisms by oncogenic DNA viruses.

#### 1.3.1. cGAS

cGAS is an enzyme that catalyzes the synthesis of the second messenger 2′3′-cyclic GMP-AMP (cGAMP), using adenosine triphosphate (ATP) and guanosine triphosphate (GTP) as substrates, as a response to cytosolic DNA sensing [91]. In order to effectively bind to DNA, nuclear cGAS must translocate to the cytosol. The nuclear export of cGAS is mediated by chromosome region maintenance 1 (CRM1) protein. cGAS carries a nuclear export signal (NES) consisting of 6 amino acids, namely LEKLKL. When this NES is mutated, cGAS is unable to move to the cytosol and cannot exert its DNA-sensing function [92]. However, during a normal response towards cytosolic DNA, the negatively charged DNA ionically interacts with the positively charged cGAS. This binding promotes a phase transition in the cytosol to liquid-like droplets and it is stimulated by the presence of zinc ions. Enzymes highly concentrate into these droplets and induce the production of cGAMP, thus, acting as microreactors. The intensity of the induction of the phase separation into liquid-like droplets correlates with the length of the DNA and the number of binding sites in it [93]. More specifically, in the presence of cytosolic DNA, the G3Bp1 stress granule engages a primary condensed state of cGAS through liquid–liquid phase separation (LLPS), inducing its activation. To continue with the activation of the cGAS/STING pathway, G3Bp1 has to dissociate from cGAS. This process can only be achieved in the presence of DNA, since RNA does not induce this dissociation [94].

Recognition of self-DNA can lead to overactivation of the innate immune system and the development of autoimmune and inflammatory diseases [95]. Thus, a tight regulation of cGAS/DNA binding is required. In resting conditions, the N-terminus of cGAS, which is responsible for the formation of liquid-like droplets and consequent activation, can be associated with the plasma membrane lipid phosphatidylinositol 4,5-biphosphate (PI(4,5)P2). Therefore, plasma membrane bound cGAS activity is suppressed, providing a mechanism to avoid interactions with self-DNA both by physical distance and also through the capturing of its biologically active domain [96]. An intact nuclear envelope provides physical separation between cGAS and nuclear DNA to avoid self-DNA targeting and the activation of an abnormal immune response. Nuclear envelope integrity can be compromised in normal physiological processes such as mitosis and post-mitotic nuclear assembly, therefore, additional regulatory mechanisms are required to prevent the binding of cGAS to self-DNA. One of these mechanisms is the chromatin-binding protein barrier-to-autointegration factor 1 (BAF). BAF dynamically displaces bound cGAS from dsDNA. Briefly, when nuclear integrity is lost, cytosolic-derived cGAS accumulates on chromatin at the nucleus but is outcompeted by BAF, which in turn binds more efficiently dsDNA. This prevents the activation of the cGAS/STING pathway [97]. If BAF is bypassed, another level of regulation comes through the interactions of cGAS and chromatin. cGAS interacts with the acidic patch and nucleosomal DNA contacts of chromatin. Specifically, H2A/H2B heterodimers and nucleosomal DNA capture the C-terminus of cGAS, rendering it unable to bind to dsDNA nor to oligomerize and activate its catalytic activity, thereby preventing innate immune activation by self-DNA [98].

Despite a correct targeting of foreign DNA, cGAS requires regulatory mechanisms to prevent its overexpression and, consequently, the overstimulation of the innate immune system. The small ubiquitin-like modifier (SUMO) proteins act as regulators of cGAS activity. The SUMO modification of molecules is known as SUMOylation and for the case of cGAS, it is catalyzed by the ubiquitin ligase TRIpartite Motif containing 38 (TRIM38) [99]. SUMOylation of cGAS enhances its molecular stability and suppresses its oligomerization, DNA-binding and transferase functions [99,100]. To counteract SUMO effects and allow for cGAS activation, the SUMO-specific peptidase 7 (SENP7) dissociates SUMO from cGAS. This process referred to as de-SUMOylation, releases cGAS and allows for the consequent activation of the TANK-binding kinase 1 (TBK1) and the IFN regulatory factor 3 (IRF3). SENP7 mediated de-SUMOylation of cGAS is critical for antiviral responses against DNA viruses, such as Herpes simplex virus 1 (HSV-1), since animals deficient in SENP7 are susceptible to HSV-1 infection accompanied with high mortality rates while these effects are not seen with RNA virus infections [100]. cGAS activity can also be blocked by increased protein degradation. K48-linked ubiquitination of cGAS at the residue K414 tags cGAS for p62-dependent autophagic degradation. During a DNA virus infection, the E3 ubiquitin-substrate ligase TRIM14, an IFN-stimulated gene (ISG), induces the cleavage of cGAS by the deubiquitinase ubiquitin-specific peptidase 14 (USP14). This inhibits cGAS degradation, allowing for further induction of the STING pathway [101]. Another deubiquitinase involved in the stabilization of cGAS is USP27X. Like USP14, USP27X inhibits the K48-linked ubiquitination of cGAS. This correlates with cGAMP levels since in cells in which USP27X is abrogated, cGAMP expression decreases. Additionally, the activation of USP27X induces cGAMP-dependent production of IFN-β in response to cytosolic DNA [102]. The covalent attachment of palmitic acid, also known as palmitoylation, to cGAS has been shown to inhibit its activity. The acyltransferase ZDHHC18 palmitoylates cGAS at C474 and this thwarts its ability to bind DNA. Additionally, cGAS dimerization is reduced and the subsequent phosphorylation and activation of STING is suspended. The inhibitory effect of cGAS palmitoylation in antiviral responses has been observed in vivo where mice deficient in ZDHHC18 are resistant to the infection of HSV-1 [103]. cGAS activity can also be inhibited by the enzymatic activity of the tubulin tyrosine ligase-like (TTLL) family of glutamylases. In this sense, the addition of polyglutamic residues at E272 by TTLL6 blocks the DNA-binding capacity of cGAS [104]. The STING-mediated production of IFN-β in response to cytosolic DNA can also be inhibited by the activation of the Aim2 inflammasome. Aim2-mediated activation of caspase 1 stimulates the expression of the pore-forming protein gasdermin D, thereby generating K+ efflux from the cell and affecting the capacity of cGAS to produce cGAMP. The subsequent decline of cGAMP production results in decreased activation of the STING/TBK1/IRF3 pathway [105]. cGAS activation is also a target for immunoevasion in cancer. The microRNA miR-23a/b, which can be found overexpressed in cancer [106], targets the 3′-UTR of cGAS mRNA. This targeting alters cGAS expression postranscriptionally since its protein levels are reduced whereas its downstream targets (TBK1, STING and IRF3) mRNA levels remain unaltered. Upregulation of miR-23a/b suppresses the phosphorylation of TBK1 as well as the nuclear translocation and phosphorylation/dimerization of IRF3. Thus, miR-23a/b acts as an inhibitor of the cGAS-STING pathway, suppressing STING-mediated immunosurveillance [107].

#### 1.3.2. cGAMP

Activated cGAS catalyzes the synthesis of the second messenger cGAMP. More particularly, the binding of DNA to cGAS induces a conformational change in the enzyme that allows it to modify GTP and ATP into the cyclic nucleotide cGAMP [89,91,108]. cGAMP can be secreted extracellularly, for instance, by cancer cells, and acts as an immunotransmitter. cGAMP-sensing cells from the innate immune system, such as monocytes, are endowed with proteins like the solute carrier family 46 member 2 (SLC46A2), which allow for the internalization of cGAMP [109]. Additionally, within the plasma membrane, there is a STING isoform that senses extracellular cGAMP. More specifically, its C-terminus is found outside of the cell and directly senses extracellular cGAMP. Extracellular cGAMP activates STING, which in turn enhances TBK1/IFN3 signalling [110]. The sensing of extracellular cGAMP can be halted by the expression of the enzyme ectonucleotide pyrophosphatase/phosphodiesterase family member 1 (ENPP1) by cancer cells. ENPP1 is localized in the plasma membrane and hydrolyzes cGAMP, thus inhibiting further activation of the STING pathway. High ENPP1 expression in cancer is associated with higher rates of metastasis and resistance to immunotherapy, namely immune checkpoint inhibitors [111].

#### 1.3.3. STING

Once synthesized, cGAMP binds with high affinity to STING, a transmembrane protein found in the endoplasmic reticulum (ER). This binding to STING induces conformational changes in the protein structure that allow for its further activation and translocation from ER to the Golgi apparatus [112]. This mechanism is necessary for the interaction between STING and TBK1, which allows for the activation of TBK1 [113]. The metabolic activity within the ER induces the production of reactive oxygen species (ROS) that have a direct negative impact on STING trafficking. Oxidative stress causes lipid peroxidation whose products 4-hydroxynonenal (4-HNE) and malondialdehyde (MDA) increase during DNA virus infection. These products can directly interact with the STING protein and, particularly 4-HNE, induce STING carbonylation. This modification traps STING within the ER and impedes its trafficking to the Golgi apparatus. The enzyme glutathione peroxidase 4 (GPX4) counteracts the release of ROS and lipid peroxidation and has been shown to be crucial for STING translocation towards the Golgi apparatus [114]. Additionally, the K63-linked polyubiquitination of STING at K224/20/289 is also necessary for the translocation of STING from ER to Golgi [115]. The ER-bound epidermal growth factor receptor (EGFR) phosphorylates STING at Y245 and prevents it from translocating to the ER-Golgi intermediate compartment (ERGIC) for autophagosomal degradation. EGFR-mediated phosphorylation of STING is required for its trafficking and the activation of IRF3 signaling. and does not alter NF-κB expression, transcription factors that induce the expression of IFNs and proinflammatory cytokines, respectively [108,116]. Mice exposed to DNA virus infection after treatment with the EGFR inhibitor gefitinib or that carried efgr−/− monocytes could not produce IFN-mediated responses and quickly succumbed to the infection [117]. The phosphorylation-dependent activation of STING is a target for immune evasion by DNA viruses. DNA virus infection induces the expression of the catalytic subunit of the protein phosphatase 6 (PPP6C). PPP6C de-phosphorylates STING, therefore preventing its translocation and activation [118]. If STING activation is not altered, the Yip1 domain family member 5 (YIPF5) recruits STING to coat protein complex II (COPII)-coated vesicles present in the ER exit sites (ERES). This promotes the DNA-induced translocation of STING to perinuclear puncta and further stimulates this pathway [119]. The accumulation of the phospholipid phosphatidylinositol 3-phosphate (PtdIns3P) increases the rate of STING trafficking from the ER to the Golgi apparatus. This, in turn, activates STING dimerization and IRF3 phosphorylation. This process is inhibited by the protein phosphatases myotubularin-related protein (MTMR) 3 and MTMR4, which de-phosphorylate PtdIns3P, suppressing its effects on STING trafficking and thereby diminishing IFN-related innate immune responses [120]. Once it has bound cGAMP, STING translocates to the ERGIC. The ERGIC vesicles that enclose STING possibly induce LC3 lipidation through a nonconventional autophagy mechanism. This induction of autophagy promotes the clearance of cytosolic DNA. This mechanism is independent on the activation of TBK1 and the C-terminal signaling of STING.

Sulfated glycosaminoglycans (sGAGs) are synthetized in the Golgi apparatus and their negatively charged sulfate groups allow for electrostatic interactions with proteins. After its translocation to the Golgi apparatus, STING binds to sGAGs through its positively charged residues. This interaction induces the polymerization of STING and its consequent activation. TBK1 is then recruited to the Golgi apparatus where it is activated and further induces IRF3 expression. The interaction of STING with sGAGs is necessary for innate immune responses against DNA viruses in vivo since mice deficient in sGAG production were more susceptible to vaccinia virus (VSV) infection compared to their wild-type counterparts [121]. Within the Golgi apparatus, STING localizes at the trans-Golgi network (TGN) domain where it undergoes palmitoylation at C88/91. This posttranslational modification promotes the clustering of STING at the TGN, which in turn promotes the closer approach of TBK1 and IRF3. This activates IFN-mediated antiviral responses, which are abrogated in cells with single amino acid substitutions at C88/91 that impede the palmitoylation of STING [122]. Another posttranslational modification of STING, SUMOylation at its residue K337 by TRIM38 promotes its oligomerization and recruitment of IRF3 while also preventing its lysosomal degradation by the chaperone-mediated autophagy (CMA) pathway [99].

STING activation requires regulatory mechanisms to prevent overstimulation or responses against self-DNA. The NOD-like receptor family member X1 (NLRX1) present in the mitochondrial outer membrane, can bind to STING as a form of negative regulation. The presence of NLRX1, impedes the formation of the STING/TBK1 complex. This, in turn, suppresses the DNA-induced activation of STING and further activation of IFN genes. Nlrx1−/− mice are resistant to infection by DNA viruses and show a sustained activation of innate immunity [123]. cGAMP is not only involved in STING activation but also acts as a negative regulator by de-phosphorylating the AMP-activated protein kinase (AMPK) at T172, which activates the unc-51 like autophagy activating kinase (ULK1). Subsequently, ULK1 phosphorylates STING at S366, which inhibits its activation and trafficking, suppressing IRF3 function [124]. It has been proposed that ULK1 phosphorylation of STING recruits the tumor necrosis factor receptor-associated factor 6 (TRAF6). TRAF6 is involved in STING-mediated activation of NF-κB rather than IRF3 in response to dsDNA. The IKK complex is composed of two catalytic subunits, IKKα (IKK1) and IKKβ (IKK2), and a regulatory subunit, IKKγ (NEMO). This complex is responsible for phosphorylating IκB, leading to its ubiquitination and subsequent degradation. This releases NF-κB, allowing it to translocate to the nucleus, where it binds to specific DNA sequences and activates the transcription of target genes [125]. This process is mediated by TBK1 that is downstream of TRAF6 and activates NF-κBp65 through the IKKαβ activation loop [126]. K48-linked polyubiquitination of STING has an inhibitory effect on its function because it targets it for proteasomal degradation. Viral infection stimulates the expression of the E3 protein-ubiquitin ligase ring finger protein 5 (RNF5). RNF5 targets STING and catalyzes its K48-linked ubiquitination tagging it for proteasomal degradation, thus, inhibiting a STING-mediated antiviral cellular response [127]. The ovarian tumor deubiquitinase 5 (OTUD5) cleaves the K48-linked polyubiquitin chains at the residue K347, preventing STING degradation and stabilizing the molecule. The presence of OTUD5 positively regulates the phosphorylation of TBK1 and IRF3 induced by cytosolic DNA [128]. Upon DNA-virus infection, the increased production of cGAMP induces the condensation of STING within the ER through a process of phase separation. STING condensates have a puzzle-like structure and capture TBK1 while leaving IRF3 outside. The sequestering of TBK1 inside the STING condensates prevents its interaction with downstream components of the pathway, resulting in the inhibition of innate immune responses towards cytosolic DNA during viral infection [129]. STING is also targeted by the E3 ubiquitin-protein ligase TRIM56, which catalyzes its ubiquitination at the K50 residue. The covalent union of the K53-linked polyubiquitin chains induces STING dimerization and TBK1 recruitment, thereby enhancing the activation of IRF3 [130].

#### 1.3.4. TBK1/IRF3/NF-κB

Following cGAMP-induced oligomerization, STING translocates to the Golgi and recruits TBK1, a protein kinase, through its C-terminal PLPLRT/SD motif. TBK1 interacts with the PLPLRT/SD motif via hydrophobic interactions and hydrogen bonds and becomes activated. Activated TBK1 phosphorylates STING, which in turn induces the recruitment and activation of more TBK1. Additionally, STING is also phosphorylated by TBK1 at the pLxIS motif, and this activates the recruitment of IRF3. STING-recruited TBK1 and IRF3 closely interact, and this facilitates the activation of IRF3 [131,132]. Upon activation, IRF3 translocates to the nucleus and binds to the promoters of Type I and III IFN genes, inducing their expression and resulting in the activation of the cellular antiviral state. Additionally, TBK1 phosphorylates the core m6A methyltransferase METTL3 at S67 to enhance its catalytic activity. METTL3 then catalyzes a m6A RNA modification on IRF3 mRNA, which promotes protein translation. TBK1-induced METTL3 expression is required for the stable expression of IFNs during DNA-virus [133]. During HSV-1 infection, cytosolic DNA-induced STING activates the signal transducer and activator of transcription 6 (STAT6). Specifically, STING-activated TBK1 phosphorylates STAT6, which in turn activates it. STAT6 is required for antiviral innate immune responses since Stat6−/− mice had a higher susceptibility to HSV-1 and VSV infection compared to wild-type mice [134]. The endoprotease caspase 3 (Casp3) is activated during apoptosis by both extrinsic and intrinsic pathways. Casp3 exerts a negative regulatory role in the activation of the cGAS/STING/IRF3 pathway. Specifically, Casp3 cleaves cGAS at D319 and IRF3 at D121/125, thus inactivating their activity. This mechanism prevents the stimulation of an innate immune response towards self-DNA during apoptosis to prevent immunogenic cell death. Additionally, virus infection can trigger the expression of Casp3; therefore, Casp3 cleavage of cGAS and IRF3 can be used as an immune evasion mechanism [135]. Along with the induction of STING, the presence of viral or intracellular DNA induces the expression of the metalloprotease Myb-like, SWIRM, and MPN domains 1 protein (MYSM1). MYSM1 removes the K63 ubiquitin chains of STING at its K150 residue. This, in turn, downregulates TBK1 and IRF3 expression [136].

The efficient activation of the immune response is key in the battle between viral clearance and viral persistence. DNA oncoviruses have developed various strategies to evade the immune system, paving the way for persistent infections. In rare cases, these mechanisms can lead to uncontrolled cellular proliferation, accumulation of mutations and subsequent cellular transformation. In particular, accumulating evidence suggests that innate immunity evasion plays a pivotal role in viral-induced carcinogenesis.

The cGAS-STING pathway, responsible for detecting foreign DNA, demonstrates a broad spectrum of antiviral activity, playing a crucial role in the host’s defense against diverse DNA viruses. Consequently, inhibiting this pathway could prove critical for DNA oncovirus replication, persistence and subsequent tumorigenesis.

## 2. Mechanisms of STING-Mediated Immune Evasion by Oncogenic DNA Viruses

Oncogenic DNA viruses have developed numerous mechanisms to efficiently thwart the cGAS-STING pathway, ensuring their replication and persistence within the host cell (Figure 1).

### 2.1. Shielding the Viral Genome from cGAS Sensing

The interaction between HBV and the innate immune system is a complex process that remains elusive and controversial. HBV is considered a “stealth” virus that does not at all or only marginally induces an IFN response in infected hepatocytes, as observed in vitro and in vivo as well as in acutely infected patients [137,138,139,140,141,142,143,144,145]. However, the mechanisms behind this lack of innate immune responses are not fully understood. In particular, studies have demonstrated that hepatocytes express low levels of STING, rendering them deficient in foreign DNA sensing machinery and contributing to the lack of IFN responses in HBV-infected hepatocytes [145,146]. In the same line, introduction of exogenous cGAS and STING in hepatocyte cell lines has been shown to enable foreign DNA sensing, resulting in restricted HBV replication, therefore suggesting that HBV does not actively suppress the cGAS-STING pathway [145,146,147,148]. Consequently, it was hypothesized that the lack of IFN responses in HBV infection would be due to poorly active STING-dependent DNA sensing mechanisms in hepatocytes instead of HBV inhibition of the host innate immune sensing functions, which may explain why HBV has adapted to specifically replicate in hepatocytes. Contrary to this, other studies demonstrated that, even when expressed at a low level in hepatocytes, the cGAS-STING pathway retains the capability to inhibit HBV replication upon activation [149,150,151,152]. Notably, two recent studies showed that naked HBV DNA is sensed by the cGAS-STING pathway, whereas the packaged HBV genome appears not to be recognized during viral infection of human hepatocytes [151,152]. During the entry process, HBV DNA is transported inside the nucleocapsid into the nucleus, where it persists as a covalently closed circular DNA and serves as a template for the transcription of the pregenomic RNA. Viral DNA synthesis later occurs via the reverse transcription of the RNA pregenome within the nucleocapsid in the cytoplasm [67,153]. This unique replication strategy may thus enable HBV to evade the detection of cytoplasmatic DNA sensors, such as cGAS, by hiding viral DNA and replication intermediates inside nucleocapsids [151,152]. This hypothesis was previously suggested in an immortalized mouse hepatocyte cell line model where destabilized HBV capsids allowed for viral DNA to readily prompt a STING-mediated response [154]. In conclusion, the shielding of HBV DNA in the viral capsids, coupled with low expression levels of the cGAS-STING pathway in hepatocytes, may be the reason for the lack of hepatic IFN response after HBV infection [144,151].

A similar strategy of shielding the viral DNA from cytosolic cGAS-STING surveillance during transit to the host cell nucleus has been identified in high-risk HPV16 [155]. The nonenveloped HPV capsid is composed of two proteins: the major protein L1 and the minor protein L2. The latter is responsible for the intracellular transport and nuclear accumulation of the viral DNA genome during infection [156]. After entry of the HPV virion through endocytosis, most of the L1 capsid is disassembled and degraded, while L2 remains associated with the viral DNA and inserts itself into endo/lysosomal membranes [157]. While only a small portion of the N-terminus of the transmembranous L2 protein remains luminal in complex with the viral genome, the cytosolic portion interacts directly with cellular trafficking factors, facilitating trafficking of the vesicular L2/DNA complex to the lumen of the TGN, where it resides during interphase [158]. Upon entry into mitosis, vesicular L2/DNA traffics away from the fragmenting Golgi and accumulates on metaphase chromosomes [50]. Analysis of the cGAS-STING response to initial HPV infection of keratinocytes showed that HPV DNA transfection resulted in acute cGAS-STING activation and downstream IFN production whereas viral DNA delivered through HPV infection elicited minimal cGAS-STING and IFN responses [155]. Furthermore, the use of cationic lipids to cause premature disruption of intracellular vesicular membranes during infection resulted in activation of the cGAS-STING pathway. These results suggest that HPV is indeed a stealthy virus, capable of evading cellular cGAS/STING surveillance thanks to its unique vesicular trafficking [155].

### 2.2. Transcriptional and Post-Transcriptional Inhibition of cGAS-STING Pathway Gene Expression

Downregulating the cGAS-STING pathway gene expression seems to be a common evasion strategy employed by oncogenic papillomaviruses. Several studies have shown a significant reduction in protein levels of STING, cGAS and TBK1 in cells infected with high-risk HPV16 and HPV18, as well as BPV2 and BPV13 [159,160,161,162]. This reduction appears to be primarily caused by downregulation at the transcription level. Oncogenic papillomavirus infection has been demonstrated to correlate with a significant decrease in the mRNA levels of STING, cGAS and TBK1 [159,161]. In particular, HPV16 and HPV18 E2, HPV18 E7 and BPV2 and BPV13 E5 proteins have all been shown to potently reduce STING mRNA levels [159,161,162,163]. One of the possible mechanisms whereby oncogenic papillomaviruses suppress STING and cGAS gene transcription was shown to be mediated by the HPV18 E7 oncoprotein. This oncoprotein upregulates the host chromatin repressor SUV39H1, which then promotes epigenetic silencing of cGAS and STING genes in HPV-transformed cells [162].

Nonetheless, it is highly probable that all oncogenic papillomaviruses do not employ identical mechanisms to suppress the expression of cGAS-STING pathway genes. In particular, it is reported that HPV16 E7, like HPV18 E7, increased SUV39H1 protein levels [162]. Other studies have shown that HPV16 E7 had only a limited effect on the mRNA transcription of STING [160,164]. Indeed, HPV16 E7 protein was demonstrated to downregulate STING expression at a posttranscriptional level, using mechanisms distinct from those used by HPV18 E7. HPV16 E7, but not HPV18 E7, interacts with NLRX1, a host protein complex scaffold recruiting autophagy-promoting molecules, to accelerate STING turnover through an autophagy-dependent mechanism [160].

Interestingly, low-risk HPV8 E1, E2, E6 and E7 proteins appeared to have no effect on STING expression levels, suggesting that HPV ability to downregulate STING gene expression could be related to its pathogenicity [164].

Contradictory data exist regarding whether HBV possesses molecular mechanisms to inhibit the cGAS-STING pathway. As previously mentioned, several studies have reported that HBV does not actively suppress cGAS-STING responses [145,146,147,148,149,150,151,165]. However, other studies suggest an active inhibition of the cGAS-STING pathway by HBV. For instance, HBV infection was shown to result in a significant downregulation of cGAS protein expression and cGAS, STING and TBK1 mRNA levels in both hepatocyte culture models and human liver chimeric mice [152,166]. The downregulation of cGAS protein expression may be attributed to the HBx protein, as proposed in two recent studies [166,167]. On the one hand, in experiments based on overexpression of cGAS, STING and HBx in human hepatocellular carcinoma cell lines, HBx was demonstrated to decrease cGAS protein levels by directly binding this protein and promoting cGAS autophagy and K48-linked ubiquitination, further inhibiting cGAS-mediated pathways [167]. On the other hand, HBV infection has been shown to upregulate the expression of host histone acetyltransferase 1 (HAT1) in primary human hepatocytes and human liver chimeric mice [166,168]. HAT1 is a type B histone acetyltransferase that is responsible for acetylation of newly synthesized histones and thus plays a central role in host chromatin assembly [169]. It was demonstrated that HBx interacts with the transcription factor Sp1 to upregulate HAT1 expression in HBV-infected cells [168]. HAT1 would, in turn, increase the levels of the miRNA miR-181a-5p by modulating acetylation in the miR-181a-5p promoter [166]. Subsequently, miR-181a-5p binds to the cGAS mRNA 3′UTR, leading to a decrease in cGAS mRNA and protein levels. Additionally, nuclear expression and localization of cGAS were found to be increased in primary and immortalized human hepatocytes and human liver chimeric mice [166]. Moreover, HBV-induced HAT1 was shown to promote the expression of karyopherin 2 (KPNA2), a host factor involved in nuclear import of cGAS [170]. In conclusion, the viral protein HBx appears to be involved directly and indirectly through HAT1 in both the transcriptional and posttranscriptional inhibition of cGAS expression, thus actively impairing the cGAS-STING pathway and IFN signaling [166,167].

Furthermore, HBV was reported to interfere with the cGAS-STING responses of immune cells that sense HBV infection in vivo. First, STING protein levels in NK cells were found to be significantly decreased in CHB patients, impairing NK cell DNA-sensing and degranulation [171]. The STING expression level of NK cells was negatively associated with serum HBsAg level. The presence of the major HBV envelope polypeptide HBsAg was suggested to inhibit STING expression and signal by inactivation of STAT3, a positive transcription factor that directly binds to the promoter of STING [171]. In the same line, studies reported that mRNAs levels of STING in peripheral blood mononuclear cells were significantly decreased in CHB patients in comparison to healthy controls. The downregulation of STING gene expression in CHB patients was attributed to the hypermethylation of the STING promoter [172].

To our knowledge, there have been no reports of downregulation of cGAS-STING pathway gene expression at the transcriptional level by oncogenic herpesviruses. However, post-transcriptional inhibition of STING expression has been demonstrated in EBV-infected human airway epithelial cells [173]. EBV was shown to upregulate the expression of the host E3 ubiquitin ligase TRIM29, a member of the TRIM family of E3-protein ligases. TRIM29 then interacts with the c-di-GMP-binding domain of STING and induces its ubiquitination at K370 site by K48-mediated linkage tagging it for proteasomal degradation [173].

### 2.3. Inhibition of cGAS DNA Binding and Activation

KSHV infection induces the activation of a cGAS-dependent innate immune response [174,175]. To circumvent this antiviral response, KSHV expresses the tegument protein ORF52 also termed KSHV inhibitor of cGAS (KicGAS). KicGAS selectively binds DNA and cGAS, thus, inhibiting the DNA-dependent activation of cGAS and, consequently, the STING pathway [174]. Specifically, KicGAS self-oligomerizes and forms liquid-like droplets upon DNA-binding that interfere with the process of phase-separation, which is critical for cGAS activation [176]. Notably, KicGAS homologs in EBV and MHV-68 similarly bind to both DNA and cGAS and inhibit cGAS enzymatic activity, suggesting an evolutionarily conserved mechanism for the inhibition of cGAS within gammaherpesviruses. Another mechanism whereby KSHV can bypass cGAS and progress to its lytic replication cycle is through the expression of the KSHV-encoded LANA [174]. The N-terminus of cytosolic LANA binds with high affinity to cGAS. This binding sequesters cGAS and impedes its activation and further phosphorylation of TBK1 and IRF3, therefore inactivating STING-mediated IFN responses.

The cGAS inhibitory effects of BAF can be hijacked by the oncogenic gammaherpesviruses KSHV and EBV. By outcompeting cGAS in dsDNA binding, BAF inhibits the STING pathway, making the host cell permissive to viral infection. Additionally, BAF was shown to promote the proteasomal degradation of cGAS in infected cells. BAF downregulation in vitro correlates with lower viral titers as well as lower viral lytic protein expression in KSHV infection. The immunosuppressive effect of BAF can be seen in primary and latently infected cells as well as in lytically reactivated cells. BAF expression is also increased in EBV primary infection and reactivation. Its abrogation correlates with higher expression of IFN genes, confirming BAF’s role in EBV-mediated immunosuppression [177].

KSHV can take advantage of another host mechanism aimed at the regulation of cGAS/STING caspase activation. Specifically, KSHV infection induces the expression of caspase 8, which in turn inhibits the activity of cGAS. Protein transcription and caspase cleavage analyses reveal that caspase 8 does not cleave cGAS directly, but potentially an upstream regulatory factor. This caspase 8-mediated downregulation of cGAS activity allows for viral reactivation and progress to the lytic cycle, which is halted by the use of caspase inhibitors. Interestingly, when IFN signaling is blocked, the effect of caspase inhibition upon lytic gene expression is rescued. This suggests that type I IFN secretion has a paracrine/autocrine effect on the inhibition of the lytic cycle mediated by caspase blockage. It is worth noting that the caspase 8 effect on viral progression does not induce apoptosis [178].

Notably, as of now, there have been no reports of direct inhibition of cGAS enzymatic activity by papillomavirus and hepatitis B virus proteins.

### 2.4. STING Inhibition through Direct Binding

Many DNA oncoviruses have been shown to evade the cGAS-STING pathway through the direct binding of viral oncoproteins to STING, thereby inhibiting STING function and downstream signaling. Two oncoproteins of KSHV were demonstrated to mechanically interact with STING. First, viral IFN regulatory factor 1 (vIRF1), a protein unique to KSHV with no homolog in other human herpesviruses, binds to STING through multiple domains and consequently disrupts the TBK1-STING interaction, preventing STING phosphorylation and activation [179]. Propionylation of lysine residues within the vIRF1 C-terminal IRF interaction domain (IAD) was determined to be necessary for the binding of vIRF1 to STING [180]. Furthermore, vIRF1 binding to STING was shown to inhibit STING-triggered IRF3 activation but not NF-κB activation [181]. Additionally, the KSHV tegument protein ORF33 interacts with the CBD domain of STING, which binds to c-di-GMP and regulates the dimerization of STING, and recruits the host protein phosphatase Mg^2+^/Mn^2+^ dependent 1G (PPM1G) to dephosphorylate p-STING and subsequently impair the recruitment of downstream IRF3 [182]. Interestingly, homologs of ORF33 in other oncogenic and non-oncogenic herpesviruses (EBV, HSV-1 and Human Cytomegalovirus) have all been found to inhibit IFN-β production, indicating that this function of ORF33 is conserved across herpesviruses.

A similar evasion mechanism was identified in the highly pathogenic and oncogenic chicken herpesvirus MDV. MDV major oncoprotein Meq was shown to bind to both STING and IRF7, impairing the assembly of the STING-TBK1-IRF7 complex and subsequently preventing the activation of TBK1 and IRF7 [183]. The C-terminal transactivation domain of Meq was found to interact with STING, while both the N-terminal bZIP domain and the C-terminal transactivation domain of Meq interact with IRF7. Chickens are IRF3-deficient, and the transcription of IFN-β in chickens is dependent on the binding of IRF7 and NF-κB transcription factors to the IFN-β promoter [184,185]. Similarly to the observations made with KSHV, MDV Meq was found to inhibit the activation of IRF7, but not that of NF-κB [183]. Meq-deficient MDV induced significantly higher levels of IFN-β in chickens than wild-type MDV, associated with a more robust CD8+ T cell response and a significant reduction in viral replication and virulence.

EBV also interacts directly with STING to inhibit its activity. During the lytic phase of EBV infection, a large tegument protein named BPLF1, which exhibits deubiquitinase activity, is expressed [186,187]. BPLF1 was found to bind directly to STING and remove all types of ubiquitin moieties on the molecule [188]. The absence of K63-linked ubiquitination would suppress STING activation and downstream IFN production [115]. Murine gammaherpesvirus 68 (MHV-68) is genetically and biologically related to EBV and KSHV and is considered an important experimental system to study virus-host interactions and viral pathogenesis [189]. MHV-68 was also found to antagonize the STING pathway through a mechanism dependent on the deubiquitinase activity of the viral protein ORF64, a homolog of BPLF1 [190]. This suggests that the deubiquitination of STING could be a common evasion strategy employed by gammaherpesviruses.

Alteration of the K63-linked ubiquitination of STING was also observed in HBV-infected cells. In human hepatic cell lines overexpressing STING, the HBV polymerase Pol was shown to specifically bind to STING through its reverse transcriptase and RNase H domains and consequently impair K63-linked ubiquitination of STING, thereby leading to a weakened IFN-β production and antiviral response [191].

Finally, direct binding-mediated antagonism of STING has also been reported in oncogenic papillomaviruses. A study showed that the oncoprotein E7 of high-risk HPV18 antagonizes the cGAS-STING pathway by direct binding of STING and further highlighted that the LXCXE motif of E7 is important for this blockade [192]. It was also shown that HPV18 E7 binds to STING in a unique region critical for NF-κB activation and blocks the nuclear accumulation of p65, thereby selectively inhibiting NF-κB signaling but not IRF3 signaling [181]. This finding differs from the IFN downregulating effect that KSHV and MDV proteins exhibit.

Notably, the oncoprotein E7 of high-risk HPV16, which shares a low degree of homology with HPV18 E7, is also responsible for potently suppressing STING-induced immune activation, but does not directly interact with STING [160]. Instead, HPV16 E5 was demonstrated to directly bind to STING and subsequently inhibit downstream signaling [193]. However, the exact mechanism via, which HPV16 E5 antagonizes STING is yet to be fully elucidated. A similar function has been identified for the oncoprotein E5 of BPV2 and BPV13 [159]. In BPV-infected cells, the formation of a ternary complex composed of E5/STING/IFI16 blocks the interaction of IFI16 with STING, and this negatively regulates the cGAS-STING signaling pathway.

### 2.5. Inhibition of TBK1 and cGAS-STING-Mediated Activation of Transcription Factors

Several DNA oncoviruses encode proteins that directly target TBK1 activity, hindering TBK1-mediated IFN production. Specifically, the tegument protein ORF11 of the gammaherpesvirus model MHV-68 has been demonstrated to interact directly with the kinase domain of TBK1 through its central domain. This results in the outcompetition of IRF3 for TBK1 binding and subsequent inhibition of TBK1-induced IRF3 activation [194]. EBV large tegument protein BPLF1, which was found to disrupt STING activity through deubiquitination, also removes all types of ubiquitin moieties on TBK1, leading to inactivation of the kinase and inhibition of TBK1-induced IRF3 dimerization [188]. In addition, increased levels of HPV16 E7 have been associated with a decrease in STING-dependent phosphorylation of TBK1 [160].

Finally, DNA oncoviruses may evade cGAS-STING immunity by impeding cGAS-STING-mediated transcription factor activation, as observed in the case of the chicken alphaherpesvirus MDV. The integral capsid protein VP23 of MDV was demonstrated to disrupt TBK1-IRF7 binding by interacting with the same region of IRF7 as TBK1, leading to the inhibition of IRF7 phosphorylation and nuclear translocation [195]. This reveals a functional redundancy between MDV VP23 and Meq, with Meq shown to impair the activation of IRF7 through direct binding to both STING and IRF7 [183] (See Section 2.4). Notably, both MDV proteins were found to inhibit the activation of IRF7, but not that of NF-κB. Nevertheless, MDV also specifically suppresses cGAS-STING-induced NF-κB activation through the action of RLORF4, an MDV-specific protein directly involved in viral virulence [196]. RLORF4 directly binds to the Rel homology domains of the NF-κB subunits p65 and p50, impeding their translocation to the nucleus. The subsequent decrease in IFN beta production was shown to drastically reduce the host CD8+ T cell response and thus enhance viral replication in vivo [196].

Overall, DNA oncoviruses have evolved precise mechanisms to disrupt multiple stages of the cGAS/STING pathway (Table 2). By manipulating the activation of cGAS/STING, these viruses compromise the efficiency of the innate immune response, creating an optimal environment for their replication and persistence. In particular, persistent infection can induce mutagenesis in the host cell, leading to neoplastic/malignant transformation. Consequently, a thorough comprehension and targeted intervention of these immune evasion mechanisms are of paramount importance to hinder oncogenic processes.

## 3. Targeting the cGAS/STING Associated Immune Evasion Mechanisms as a Therapeutical Approach

The cGAS/STING pathway has gained significant attention in cancer research due to its role in the immune response against cancer cells. As previously described in this review, this pathway is involved in the recognition of cytoplasmic DNA, a common feature of DNA oncoviruses and some cancer cells, triggering an innate immune response. Therefore, the STING pathway has become a promising therapeutic target for harnessing the innate immune system’s ability to recognize and eliminate aberrant/cancerous cells thereby enhancing the effects of current cancer therapeutics. In this line, current clinical trials evaluating drugs targeting the cGAS/STING pathway are primarily focused in cancers with no viral etiology [199,200,201,202].

In regard to neoplastic diseases caused by DNA oncoviruses, therapeutic interventions targeting the STING pathway involve a multifaceted approach aimed at restoring or stimulating its functions to counteract the viral immune evasion mechanisms. Preclinical studies have brought forward the positive impact of acting upon this pathway in DNA oncovirus-induced cancers, particularly those caused by HPV and HBV.

Head and neck squamous cell carcinoma (HNSCC) ranks among the top 10 most prevalent cancers globally. Beyond environmental factors like alcohol and tobacco, HPV infection can also trigger HNSCC. HPV-positive (HPV+) HNSCC differs histologically and molecularly from HPV-negative HNSCC, requiring different therapeutical approaches [203]. As part of their treatment, some HNSCC patients receive cetuximab, a monoclonal antibody targeting the epidermal growth factor receptor (EGFR). Cetuximab was shown to induce favorable clinical responses by activating NK cell secretion of IFNγ and inducing dendritic cell (DC) maturation [204]. As previously detailed (Table 2), HPV inhibits IFN I secretion by disrupting the cGAS/STING pathway. Consequently, HPV+ HNSCC patients with downregulated STING exhibit a less effective immune response to cetuximab treatment [203]. Combining cetuximab with a STING agonist (a cyclic dinucleotide—CDN) enhances NK and DC activation, leading to improved cancer clearance and treatment response. This combination therapy holds promise for optimizing the efficacy of treatment in HPV+ HNSCC cases with compromised STING functionality [203]. In vivo studies in murine models of papilloma have corroborated these findings. Briefly, mice whose tumors were treated with c-di-GMP, a STING agonist, showed significantly higher tumor regression rates than their PBS-treated counterparts. Remarkably, even tumors lacking STING expression displayed responsiveness to the treatment, emphasizing the pivotal role of the immune system in mediating the observed therapeutic effects [205]. Another combinatorial use of STING agonists is along with immune checkpoint inhibitors (ICI), particularly those targeted to the programmed cell death protein 1 (PD-1) and its ligand PD-L1. Studies in mice demonstrated that the intratumoral delivery of ML-RR-CDA, another STING agonist, in HPV16+ oropharyngeal tumors sensitized them to ICIs and resulted in better regression rates than those that received only ICI treatment. These findings suggest that STING-mediated activation of the innate immune system can potentially boost the effectiveness of ICI therapy in stimulating the adaptive immune system [206].

Chronic HBV infection is the leading cause of hepatocellular carcinoma [73] (Table 1). Immune evasion mechanisms, including cGAS/STING inactivation, allow for the chronicity of the infection [73,165]. It is, therefore, of crucial importance to reestablish immune surveillance to clear the infection and prevent hepatocyte neoplastic transformation. In vitro and in vivo murine studies showed that when 5,6-dimethylxanthenone-4-acetic acid (DMXAA), a murine STING agonist, was used either on hepatocytes or on liver macrophages, HBV infection was cleared. More specifically, innate immune evasion was halted and this led to the inhibition of HBV replication mediated by IFN-I secretion [149,207]. Additional in vitro studies using three different STING agonists, cGAMP, 3′,3′-c-di(2′F,2′dAMP) and its bis(pivaloyloxymethyl) prodrug, corroborated that the activation of STING promotes viral clearance in human hepatocytes and non-parenchymal liver cells [208]. Similar effects were seen with the use of the topoisomerase II inhibitor daunorubicin, which activated the cGAS/STING pathway in vitro, suppressing HBV production [209]. In light of these findings, several STING modulators are being developed and proposed as alternatives for treating HBV infection [210].

While investigations into the therapeutic potential of STING in cancers induced by herpesviruses are currently limited, there is promise in the use of STING agonists for prophylactic infection management and potential cancer prevention. In a study involving mice, treatment with a STING agonist administered up to 72 h prior to HSV-1 infection resulted in an IFN-mediated inflammatory response, effectively preventing virus replication and infection. Notably, STING agonists exhibited superior effectiveness and generated a more localized inflammatory response compared to Toll-like receptor (TLR) agonists [211]. Although HSV-1 is not an oncogenic virus, these findings underscore the potential immunostimulatory effect of STING agonists against oncogenic herpesvirus infections (Table 1 and Table 2). Nevertheless, additional research is required to thoroughly understand the effects of STING agonists in herpesvirus-mediated cancers or active infections.

In conclusion, this paper has delved into the intricate realm of oncovirus-mediated immune escape, shedding light on the pivotal role of the STING pathway in orchestrating immune responses. Through a comprehensive exploration of the mechanisms and interactions involved, we uncovered the potential of STING as a key player in bolstering the immune defenses against DNA oncoviruses. The evolving landscape of cancer research underscores the importance of unraveling the complexities of immune escape mechanisms to pave the way for innovative therapeutic strategies. As we unmask the complex interplay between oncoviruses and the immune system, the insights obtained present new avenues for therapeutic interventions, bringing us closer to more effective strategies in the ongoing battle against oncovirus-mediated immune evasion.

## Figures and Tables

**Figure 1 viruses-16-00574-f001:**
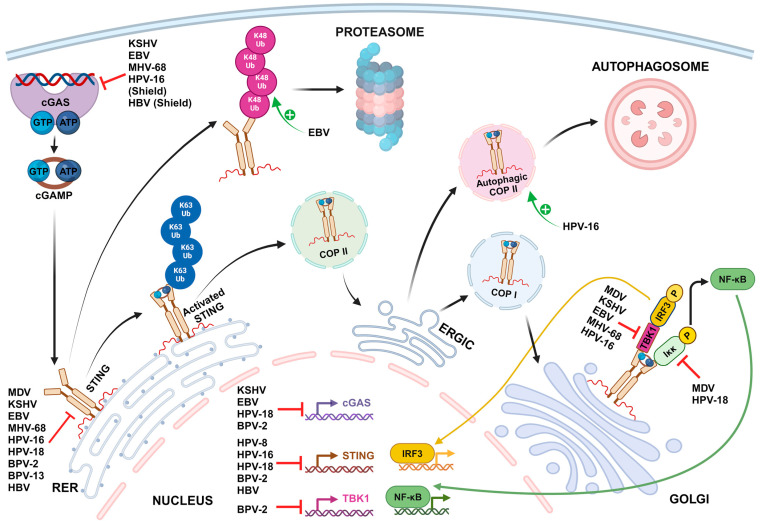
Mechanisms of STING-mediated immune evasion by oncogenic DNA viruses. DNA oncoviruses exhibit a range of sophisticated immune evasion strategies, enabling them to avoid surveillance to effectively replicate and persist within host cells. Among these evasion mechanisms, targeting the cGAS/STING pathway is instrumental in generating an inefficient response from the host’s innate immune system, thereby facilitating the progression of viral infections. Diverse families of DNA oncoviruses intervene at multiple stages of the cGAS/STING pathway, leading to its inactivation. BPV, bovine papillomavirus; EBV, Epstein Barr virus; HBV, hepatitis B virus; HPV, human papillomavirus; KSHV, Kaposi’s sarcoma-associated herpesvirus; MDV, Marek’s disease virus; MHV, murine gammaherpesvirus. Created with BioRender.com., accessed on 21 March 2024.

**Table 1 viruses-16-00574-t001:** DNA oncoviruses of relevance in human and animal health.

Family	Virus	Host	Associated Cancers	Target Cells
*Herpesviridae*	Marek’s disease virus	Chicken	T cell lymphoma	T lymphocytes
	Kaposi sarcoma-associated virus	Human	Kaposi sarcoma Primary effusion lymphoma Multicentric Castleman’s disease	B lymphocytes, Endothelial cells
	Epstein–Barr virus	Human	Burkitt’s lymphoma Hodgkin’s lymphoma Other lymphomas Nasopharyngeal carcinoma Gastric carcinoma	B lymphocytes Epithelial cells
*Papillomaviridae*	Human papillomavirus types 16, 18, 31, 33, 35, 39, 45, 51, 52, 56, 58, 59	Human	Cervical cancer Oropharyngeal carcinoma Other anogenital cancers	Skin and mucosa epithelial cells
	Bovine papillomavirus types 1, 2, 4, 13	Bovine	Urinary bladder cancer Oral squamous cell carcinoma	Skin and mucosa epithelial cells
*Hepadnaviridae*	Hepatitis B virus	Human	Hepatocellular carcinoma	Hepatocytes
*Polyomaviridae*	Merkel cell polyomavirus *	Human	Merkel cell carcinoma	Dermal fibroblasts

* To our knowledge, no study on the mechanisms of evasion of the cGAS-STING pathway by Merkel cell polyomavirus has been published so far. Consequently, this virus will not be further discussed in this review.

**Table 2 viruses-16-00574-t002:** Mechanisms of STING-mediated immune evasion by oncogenic DNA viruses.

Family	Virus	Viral Protein	Evasion Mechanism	Experimental System	References
**Shielding the viral genome from cGAS sensing**					
*Papillomaviridae*	Human papillomavirus type 16	Minor capsid protein L2	L2-mediated trafficking of viral DNA into vesicular membranes for delivery to the host cell nucleus	Immortalized human keratinocyte cell line HaCaT and Primary human foreskin keratinocytes	[155]
*Hepadnaviridae*	Hepatitis B virus	Nucleocapsid	Hiding viral DNA and replication intermediates inside the nucleocapsid	Immortalized mouse hepatocyte cell line AML12HBV10	[154]
				Immortalized human hepatocyte cell line HepG2-hNTCP and Primary human hepatocytes (PHHs)	[151,152]
**Transcriptional and post-transcriptional inhibition of cGAS-STING pathway gene expression**					
*Herpesviridae*	Epstein–Barr virus	-	Upregulation of host E3 ubiquitin ligase TRIM29, which then interacts with the c-di-GMP-binding domain of STING and induces its ubiquitination at Lys370 site by K48-mediated linkage for protein degradation	Human healthy airway epithelial cell line BEAS-2B, EBV-negative nasopharyngeal epithelial cell line NP69 and Human nasopharyngeal carcinoma cells CNE1	[173]
*Papillomaviridae*	Human papillomavirus types 16 and 18	Early protein E2	Significant reduction of STING mRNAs levels. The transactivation amino-terminal domain of E2 is involved in the suppressive effect.	Human primary keratinocytes transduced by HPV E2	[163]
	Human papillomavirus type 18	Oncoprotein E7	Transcriptional activation of the host chromatin repressor SUV39H1, which then promotes epigenetic silencing of cGAS and STING genes	Immortalized human keratinocyte cell line NIKS stably harboring a high viral load of HPV18 episomal genomes (NIKSmcHPV18 cells) and Immortalized cervical carcinoma-derived cell line harboring integrated HPV18 DNA (HeLa cells)	[161,162]
	Human papillomavirus type 16	Oncoprotein E7	Interaction with NLRX1, a host protein complex scaffold recruiting autophagy-promoting molecules, to accelerate STING turnover through an autophagy-dependent mechanism	HPV16-positive head and neck squamous cell carcinoma (HNSCC) cell lines (93VU147T, UMSCC47 and SCC90 cells) and HPV-negative HNSCC cell line FaDu HPV16 E6/E7-expressing HNSCC mouse model, MOC2-E6/E7	[160]
	Bovine papillomavirus types 2 and 13	-	Significant reduction of cGAS, STING and TBK1 mRNAs in BPV-infected cells	BPV-infected bladder mucosa samples from cows with bladder neoplasms	[159]
*Hepadnaviridae*	Hepatitis B virus	HBx	Decrease of cGAS protein levels by direct binding to cGAS and promoting cGAS autophagy and K48-linked ubiquitination	Human hepatocellular carcinoma cell lines (HEK293T, SMMC-7721 and LO2 cells) transfected with an HBx plasmid	[167]
			Sp1 upregulates the expression HAT1. HAT1 increases miRNA levels of miR-181a-5p by modulating acetylation in the miR-181a-5p promoter. MiR-181a-5p in turn binds to the cGAS mRNA 3′UTR, decreasing cGAS mRNA and protein levels.	Immortalized human hepatocyte cell lines (HepG2-hNTCP, Huh7 and HepG2 cells), PHHs and Human liver chimeric mice	[166,168]
			Sp1 upregulates the expression HAT1. HAT1 increases miRNA levels of miR-181a-5p by modulating acetylation in the miR-181a-5p promoter. MiR-181a-5p in turn binds to the cGAS mRNA 3′UTR, decreasing cGAS mRNA and protein levels.	HepG2-hNTCP, Huh7 and HepG2 cells, PHHs and Human liver chimeric mice	[166,170]
		HBsAg	Inhibition of STAT3 and subsequent downregulation of STING expression in NK cells of patients with chronic hepatitis B (CHB)	NK Cells from HBeAg-Negative CHB Patients and Human NK cell line (NK-92 cells)	[171,197]
		-	Hypermethylation of the STING gene promoter inducing significantly lower levels of STING mRNA in peripheral blood mononuclear cells (PBMCs) of CHB patients	Isolated PBMCs of CHB patients	[172,198]
**Inhibition of cGAS DNA binding and activation**					
*Herpesviridae*	Kaposi sarcoma-associated herpesvirus	Tegument protein ORF52/KicGAS	KicGAS self-oligomerizes and forms liquid droplets upon binding to DNA, thus inhibiting the DNA-induced phase separation and activation of cGAS	HEK293T cells stably expressing STING, THP1 Lucia™ ISG cells (InvivoGen), which express luciferase from a gene under the control of an IRF3-inducible promoter and Human primary lymphatic endothelial cells	[174,176]
		LANA	Direct binding of cytoplasmic isoforms of LANA to cGAS antagonizes cGAS function	Primary effusion lymphoma-derived B-cell line BCBL-1, HEK 293T and HeLa cells, HuAR2T.rKSHV.219, a conditionally immortalized endothelial cell line persistently infected with recombinant virus rKSHV.219	[175]
		-	Upregulation of barrier-to-autointegration factor 1 (BAF) expression, a host protein that induces the degradation of cGAS through the proteasomal pathway	Kaposi’s sarcoma-derived cell line SLK, iSLK.219 cell line, which is latently infected with recombinant virus rKSHV.219, TREx-BCBL1-RTA cell line, a KSHV-infected BCBL-1 cell line	[177]
		-	Activation of caspase-8 during lytic reactivation, which indirectly inhibits cGAS enzymatic activity	iSLK.219 cell line, BC3 cells, a KSHV-infected B cell line derived from a primary effusion lymphoma patient	[178]
	Epstein–Barr virus	Tegument protein BLRF2 (KSHV ORF52 homolog)	Binding to both DNA and cGAS and inhibition of cGAS enzymatic activity	HEK293T cells stably expressing STING	[174,176]
		-	Upregulation of BAF expression, a host protein that induces the degradation of cGAS through the proteasomal pathway	Human gastric adenocarcinoma-derived cell line AGS and AGS-EBV cell line latently infected with GFP-expressing recombinant EBV	[177]
	Murine gammaherpesvirus 68	Tegument protein ORF52	Binding to both DNA and cGAS and inhibition of cGAS enzymatic activity	HEK293T cells stably expressing STING	[174,176]
**STING inhibition through direct binding**					
*Herpesviridae*	Marek’s disease virus	Oncoprotein Meq	Binding to both STING and IRF7, impeding the assembly of the STING-TBK1-IRF7 complex and the subsequent activation of TBK1 and IRF7, but not that of NF-κB	Immortalized chicken fibroblast cell line DF-1, chicken embryo fibroblasts (CEFs) and chickens	[183]
	Kaposi sarcoma-associated herpesvirus	vIRF1	Direct binding to STING through multiple domains and subsequent disruption of the TBK1-STING interaction, preventing STING phosphorylation and activation	Human umbilical vein endothelial cells (HUVECs), Immortalized endothelial cells EA.hy926 and rKSHV.219 iSLK cell line	[179]
			vIRF1 promotes its own propionylation, which is required for effective binding to STING	HEK293T and EA.hy926 cell lines, KSHV infected cell line iSLK-RGB	[180]
			vIRF1 binding to STING inhibits STING-triggered IRF3 activation but not NF-κB activation	HEK293T cells	[181]
		Tegument protein ORF33	Binding to the CBD domain of STING and recruitment of the host protein phosphatase PPM1G to dephosphorylate p-STING and subsequently impair the recruitment of downstream IRF3	rKSHV.219 iSLK, THP-1 and HEK293 cell lines	[182]
	Epstein–Barr virus	Tegument protein BPLF1	Removing of all types of ubiquitin moieties on STING, therefore suppressing STING activation and recruitment of TBK1	HEK293 cells overexpressing BPLF1, cGAS and STING, HEK293-M81 cells constitutively carrying EBV M81 strain	[188]
	Murine gammaherpesvirus 68	Tegument protein ORF64 (EBV BPLF1 homolog)	Antagonism of the STING pathway through a mechanism dependent on the deubiquitinase activity of ORF64, a homolog of EBV BPLF1	Murine dendritic cells, Wild-type and STING^gt/gt^ mice	[190]
*Papillomaviridae*	Human papillomavirus type 18	Oncoprotein E7	Binding to STING inhibits NF-κB activation and p65 nuclear accumulation, but not IRF3 activation	HeLa cells overexpressing STING, Primary mouse embryonic fibroblasts transduced with retroviral expression vectors containing HPV18 E7, HEK293T cells	[181,192]
	Human papillomavirus type 16	E5	Direct binding to STING and subsequent inhibition of downstream signaling	Human HNSCC cell line CAL-27	[193]
	Bovine papillomavirus types 2 and 13	Oncoprotein E5	Formation of a ternary complex composed of E5/STING/IFI16, which blocks interaction of IFI16 with STING	BPV-infected bladder mucosa samples from cows with bladder neoplasms	[159]
*Hepadnaviridae*	Hepatitis B virus	Polymerase Pol	Direct binding to STING and subsequent disruption of its K63-linked ubiquitination	Huh7, HEK 293 and HepG2 derivative HepaAD38 cells overexpressing STING, PH5CH8 cells, differentiated proliferative human hepatoma-derived cells HepaRG and PHHs	[191]
**Inhibition of TBK1 and cGAS-STING-mediated activation of transcription factors**					
*Herpesviridae*	Marek’s disease virus	Capsid protein VP23	Interaction with IRF7 and disruption of its binding to TBK1, leading to the inhibition of IRF7 phosphorylation and nuclear translocation	CEFs, chicken macrophage HD11 cells, DF-1 cells	[195]
		RLORF4	Binding to the Rel homology domains of the NF-κB subunits p65 and p50, interrupting their translocation to the nuclei and thereby inhibiting IFN beta production.	DF-1 cells, HEK293T cells, CEFs, chickens	[196]
	Epstein–Barr virus	Tegument protein BPLF1	Removing of all types of ubiquitin moieties on TBK1, leading to inactivation of the kinase inhibition of TBK1-induced IRF3 dimerization	HEK293 cells overexpressing BPLF1, cGAS, STING, TBK1 and IRF3	[188]
	Murine gammaherpesvirus 68	Tegument protein ORF11	Direct binding to TBK1, reducing the interaction between TBK1 and IRF3 and subsequently inhibiting IRF3 activation	HEK293T cells, Murine embryonic fibroblasts and Raw264.7 macrophage cells	[194]
*Papillomaviridae*	Human papillomavirus type 16	Oncoprotein E7	Decrease in phosphorylation of TBK1 with increased levels of HPV16 E7	93VU147T, UMSCC47, and FaDu cells overexpressing HPV16 E7	[160]

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
