# Peer review of "STINGing Defenses: Unmasking the Mechanisms of DNA Oncovirus-Mediated Immune Escape"

_viruses, 2024, doi:10.3390/v16040574_

Round 1

Reviewer 1 Report

Comments and Suggestions for Authors

In this excellent review Mayra Martinez-Lopez and coworkers review the current literature for the cGAS-STING pathway with regard to the interactions between DNA oncogenic viruses and the innate immune system.

My specific comments include:

·        Lines 90-103 and 746-757 regarding chicken’s Herpesviruses can be omitted. Similarly, line 1 from Table 1 should be deleted.

·        Lines 187-205 regarding cattle Papillomaviruses can be omitted.

·        In lines 278-279 is stated that “Necrotic cells release intracellular components such as DNA, that are detected by the immune system”. Cytosolic dsDNA can be derived from multiple sources, including genomic, mitochondrial, and exogenous origins. For example, chromosomal instability is a primary source of cytosolic dsDNA. Apart from the nuclear compartment, the mitochondria may serve as another genomic source to stimulate the cytosolic dsDNA-sensing pathway in cancer. Other sources of genomic substrates, such as apoptotic derived DNA, exosomes, and transposable elements, may also elicit cGAS–STING activation in tumors. The above must be added to this paragraph.

·        In lines 279-280 is stated that “Specifically, foreign or extracellular DNA is recognized by the innate immune system through the cyclic GMP-AMP synthase (cGAS) – stimulator of interferon genes (STING) pathway, activating in turn host immunity.” Besides this, other dsDNA sensors exist such as TLR9, DNA-dependent protein kinase, RNA polymerase III, DEAD box polypeptide 41 (DDX41), PYHIN protein family, AIM-2-like receptors (ALRs) etc, which they can activate innate immunity of the host. This must be added to this paragraph.

·        The size of the text in Figure 1, it is better to be increased, in order to be more easily read.

·        The description of STING Pathway (lines 292-549) is rather too long and analytical and it could be reduced in size, as this review focuses to STING mediated immune evasion and not to the physiological role of STING Pathway.

Comments on the Quality of English Language

Minor editing of English language required

Author Response

We thank reviewer's 1 comments that we feel will significntly improve our review paper. We hereby present our point-to-point responses to Reviewer's 1 comments.

  1. Lines 90-103 and 746-757 regarding chicken’s Herpesviruses can be omitted. Similarly, line 1 from Table 1 should be deleted.
  2. Lines 187-205 regarding cattle Papillomaviruses can be omitted.

Authors' response to both comments: We appreciate the comments, however we feel that animal oncogenic viruses should remain in the text due to the importance of these pathologies in public health and economy. We also believe this information gives a broader approach to the reader.

  1. In lines 278-279 is stated that “Necrotic cells release intracellular components such as DNA, that are detected by the immune system”. Cytosolic dsDNA can be derived from multiple sources, including genomic, mitochondrial, and exogenous origins. For example, chromosomal instability is a primary source of cytosolic dsDNA. Apart from the nuclear compartment, the mitochondria may serve as another genomic source to stimulate the cytosolic dsDNA-sensing pathway in cancer. Other sources of genomic substrates, such as apoptotic derived DNA, exosomes, and transposable elements, may also elicit cGAS–STING activation in tumors. The above must be added to this paragraph.

Authors' response: We have significantly improved the text to include this excellent point made by Reviewer 1.

  1. In lines 279-280 is stated that “Specifically, foreign or extracellular DNA is recognized by the innate immune system through the cyclic GMP-AMP synthase (cGAS) – stimulator of interferon genes (STING) pathway, activating in turn host immunity.” Besides this, other dsDNA sensors exist such as TLR9, DNA-dependent protein kinase, RNA polymerase III, DEAD box polypeptide 41 (DDX41), PYHIN protein family, AIM-2-like receptors (ALRs) etc, which they can activate innate immunity of the host. This must be added to this paragraph.

Authors' response: We have changed the text to include this excellent point made by Reviewer 1.

  1. The size of the text in Figure 1, it is better to be increased, in order to be more easily read.

Authors' response: We have modified the fonts in Figure 1 in order to make the text more easily read.

  1. The description of STING Pathway (lines 292-549) is rather too long and analytical and it could be reduced in size, as this review focuses to STING mediated immune evasion and not to the physiological role of STING Pathway.

Authors' response: We have made important changes to this section to address the issue made by Reviewer 1. Among them, we have deleted several experimental data that were deviating from the main focus of the Review.

Reviewer 2 Report

Comments and Suggestions for Authors

This review is very well-written and balanced. The authors have covered the key mechanisms of DNA virus oncogenesis and role of the cellular cGAS-STING pathway. The figure and tables are accurate and appropriate.

No major concerns.

Author Response

We thank reviewer 2 for the comments. There were no major issues to be addressed in this sense.

Reviewer 3 Report

Comments and Suggestions for Authors

The review article by Martinez-Lopez et al., “STINGing Defenses: Unmasking the Mechanisms of….”, describes the role of STING and its upstream and downstream regulators in cellular response to DNA viruses, and the mechanisms that various viruses have evolved in order to circumvent this antiviral pathway.  

Overall Assessment:  This review is comprehensive and a nice place to start for individuals looking to move into the STING field or simply looking to understand how STING might play a role in the biological systems they are studying.  The focus on relevant viruses, basic mechanisms of STING activation and signaling, and then the comprehensive list of viruses and viral proteins that inhibit this response, and the stages in the pathways impacted, was nice and well organized – particularly Table 2 that allows readers to quickly home in on their favorite virus.  A few technical glitches that need to be fixed and a bit more editing in one section would benefit the overall readability of the review and improve the flow.  Those comments are below.

Specific comments:

The citation in lines 187-206 are messed up and need to be fixed.  The number of fers cited in that section align with the overall number in the references section, and so it should be an easy formatting fix.

The writing style and grammar in the short section on “Cancer Immunosurveillance” (lines 253-291) in particular, and into the next section on STING (292-550) to a lesser extent, is inconsistent with other sections and needs additional editing.  It needs a thorough grammar check and spelling check (particularly from 268-292).  Some additional narrative on cancer immunosurveillance may be needed to provide a better introduction to the next several sections, but this section in particular needs closer attention to plural versus singular statements, grammar, etc.

Immediately after that are sections on individual signaling intermediates involved in STING activation or signaling.  Recognizing that STING regulation is complex and some of the concepts not very intuitive (the “liquid-like” have been around for a while, and are still challenging to visualize), but even a few sentences on the signaling factors to introduce them better would help.  cGAS and TBK1, in particular, are discussed as if the reader has a lot of prior knowledge about them.  In both cases (and perhaps for a few others like IRF3), a few sentences placing them in context in STING signaling versus other pathways (RLH, TLR, etc.) where activation of downstream intermediates can be quite different.  At the very least, perhaps start the individual sections in which they are discussed with a generalized introduction, such as “cGAS is a (type of protein?)….that is crucial for catalyzing  (discuss cGAMP catalyzation)……in response to …”  etc., rather than just jumping in the section on cGAS by talking about its subcellular localization.  Both TBK1 and cGAS are discussed extensively with respect to how they are perturbed by these viruses, but readers without much TBK1 or cGAS prior experience will find it hard to understand how these things are unique to the pathways discussed herein. A short introduction on these proteins with the assumption that the reader has never heard of them before would help.  It need not be extensive – just a few sentences.

Overall, the paper adds to the literature and is worthy of publication. 

Comments on the Quality of English Language

My comments are made in the overall comments to the authors.  There are a few points where the grammar can be improved, but in general, I think that minor additional editing will suffice.

Author Response

We deeply appreciate reviewer's 3 constructive comments that we feel will overall improve our review paper. We hereby offer a point-to-point responses to Reviewer's 3 comments.

  • The citation in lines 187-206 are messed up and need to be fixed.  The number of fers cited in that section align with the overall number in the references section, and so it should be an easy formatting fix.

Authors' response: We thank Reviewer 3 for this excellent observation. We have corrected the references in this section to align to the document's format and made a thorough check of the references throughout the rest of the manuscript.

  • The writing style and grammar in the short section on “Cancer Immunosurveillance” (lines 253-291) in particular, and into the next section on STING (292-550) to a lesser extent, is inconsistent with other sections and needs additional editing.  It needs a thorough grammar check and spelling check (particularly from 268-292).  Some additional narrative on cancer immunosurveillance may be needed to provide a better introduction to the next several sections, but this section in particular needs closer attention to plural versus singular statements, grammar, etc.

Authors' response: We have significantly edited this section so as to be more coherent with the rest of the text and made thorough grammar and vocabulary revisions.

  • Immediately after that are sections on individual signaling intermediates involved in STING activation or signaling.  Recognizing that STING regulation is complex and some of the concepts not very intuitive (the “liquid-like” have been around for a while, and are still challenging to visualize), but even a few sentences on the signaling factors to introduce them better would help.  cGAS and TBK1, in particular, are discussed as if the reader has a lot of prior knowledge about them.  In both cases (and perhaps for a few others like IRF3), a few sentences placing them in context in STING signaling versus other pathways (RLH, TLR, etc.) where activation of downstream intermediates can be quite different.  At the very least, perhaps start the individual sections in which they are discussed with a generalized introduction, such as “cGAS is a (type of protein?)….that is crucial for catalyzing  (discuss cGAMP catalyzation)……in response to …”  etc., rather than just jumping in the section on cGAS by talking about its subcellular localization.  Both TBK1 and cGAS are discussed extensively with respect to how they are perturbed by these viruses, but readers without much TBK1 or cGAS prior experience will find it hard to understand how these things are unique to the pathways discussed herein. A short introduction on these proteins with the assumption that the reader has never heard of them before would help.  It need not be extensive – just a few sentences.

Authors' response: We have added a short overview of the role and function of the afforementioned molecules for the benefit and better understanding readers that are not experts in the field.